# A Multi-Criteria Decision Intelligence Framework to Predict Fire Danger Ratings in Underground Engineering Structures

**Muhammad Kamran** [1,*], **Waseem Chaudhry** [2], **Ridho Kresna Wattimena** [3], **Hafeezur Rehman** [4] **and Dmitriy A. Martyushev** [5]

1   School of Engineering, University of Tasmania, Hobart, TAS 7001, Australia
2   Department of Petroleum Engineering, Institute Technology of Bandung, Bandung 40132, Indonesia; wasim_ch2@hotmail.com
3   Department of Mining Engineering, Institute Technology of Bandung, Bandung 40132, Indonesia; rkw@mining.itb.ac.id
4   Department of Mining Engineering, Faculty of Engineering and Architecture, BUITEMS, Quetta 87300, Pakistan; hafeezur.rehman@buitms.edu.pk
5   Department of Oil and Gas Technologies, Perm National Research Polytechnic University, 614990 Perm, Russia; martyushevd@inbox.ru
*   Correspondence: m.kamran@utas.edu.au

**Abstract:** A wide variety of natural catastrophes are induced by coal mining, with fire hazard being one of the most significant threats to underground engineering structures. In recent years, there has been an alarming rise in mine fire accidents due to the abundance of coal deposits around the world. Underground fires and explosions have continuously been the primary reason for a significant proportion of deaths and the destruction of infrastructure over the last few decades. Underground mining fires deplete natural coal resources, have an adverse impact on the environment by releasing hazardous chemicals and greenhouse gases into the atmosphere, and cause subsidence due to coal depletion during the combustion process. This study aims to predict fire danger rating of underground mining production processes by using the application of state-of-the-art unsupervised and supervised machine learning techniques. The developed k-nearest-neighbors-based isometric feature mapping and fuzzy c-means clustering algorithm has shown its dependability and superiority with a higher accuracy and has been advantageous to the monitoring and prevention of fire danger in underground mining production processes. The proposed multi-criteria decision intelligence framework permits early fire detection, providing the emergency response team extra time to respond the critical situations in order to prevent the fire from spreading, hence promoting sustainable, green, climate-smart, environmentally friendly and safe mining engineering operations.

**Keywords:** underground fire; safety; KNN; coal; fuzzy c-means clustering algorithm; ISOMAP

## 1. Introduction

Mine fires involving coal constitute a significant risk to underground working process safety, and can result in significant financial damage. Under normal conditions, coal will ignite due to a reaction between carbon in the coal and oxygen in the air. The coal itself is usually the cause of mining fires across the world [1]. There has been an alarming increase in the occurrence of mine fires recently due to the worldwide abundance of coal deposits. Mine fires can vary greatly in both type and severity depending on location. Mine fires are an issue in countries that generate a large quantity of coal [2]. In particular, with respect to confronting fire danger in natural energy sources along with public health and safety, underground mining fires reduce natural coal deposits, having an impact upon the environment as a result of the greenhouse gases in the atmosphere and harmful chemicals, but also actually cause subsidence as a consequence of the depletion of coal during the process of combustion [3]. Underground mining is the most common method of extracting

hard coal. The bulk of the coal reserve is recovered from the mine by applying extremely efficient automated longwall formations. The output is then refined in surface operations to produce a final product that may be commercialized [4]. Figure 1 depicts the production of hard coal based on a general understanding of the procedure.

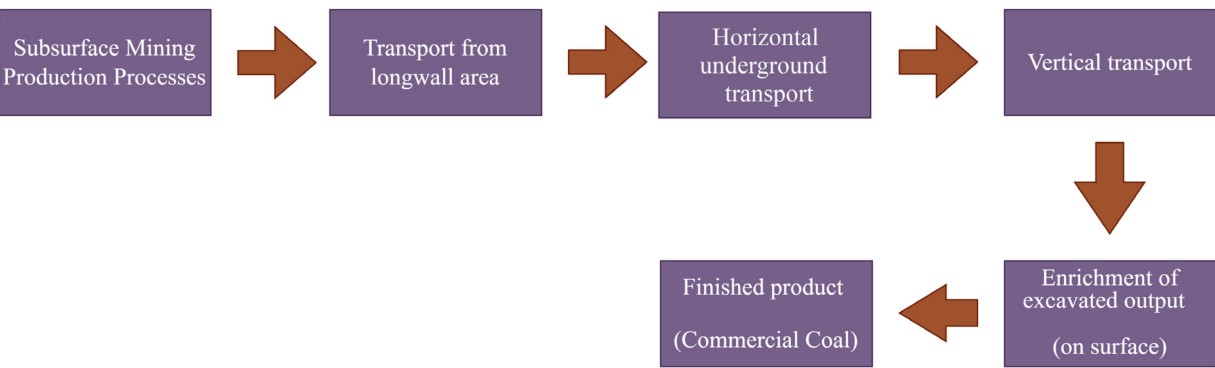

**Figure 1.** Various steps involved in the production of coal in underground engineering processes.

The process of underground mining engineering is inherently dangerous due to the uniqueness and unpredictability of the area in which it is performed. This is due to the fact that during this activity, the equilibrium in the rock mass is disturbed, which can result in a variety of environmental disasters. There are critical minerals needed for the transition to sustainable renewable resources, without which the world would not be able to achieve new climate goals. Growing mineral demand increases not just the extent of mining operations, but also their level of complexity. Sustainable mining production may become scarce if mineral supply systems are unable to keep up with market growth. Therefore, it is essential to extract critical minerals in a manner that is sustainable, commercially feasible, environmentally friendly, and socially beneficial. Several efforts have been initiated with the goal of developing sustainable, green and environmentally procedures to be used in the production process of surface and underground mining technologies. Because of this, communities that are concerned about the environment have a great deal of concern around mining industry processes [5]. Rock engineering processes have completely adhered to the principles of sustainable development by minimizing and working toward eliminating the damage induced by its activities in accordance with sustainable production strategies [6]. Environmentally friendly measures, such as "green mining" and "climate-smart mining", can be used to restore the damage that has been caused by traditional mining production processes. Utilization of environmentally friendly practices and equipment is encouraged under green mining policies. This helps to improve a mine's overall environmental performance [7]. The manufacturing techniques used in mining engineering processes can potentially generate a wide range of environmental problems. The release of hazards and the disposal of solid waste materials and discharge from mining activities onto the surface of the earth, in the absence of appropriate legislation, are the primary causes of the majority of the globe's pollution.

Fires in mines, whether in underground or above-ground mining, may cause extensive damage to the mining engineering production process; thus, researchers all around the world have tried a broad variety of strategies to mitigate this challenge. For instance, a fuzzy logic-based mine fire monitoring and control system was presented, which makes use of a wireless sensor system. This strategy involved taking readings of the immediate surroundings and sending them to a transceiver that was in constant contact with an on-the-ground control room [8]. In order to protect the mining and natural resources workforce from flames and minimize the risk of fire advancing, we must prevent the fire from spreading. Bhattacharjee, Roy [9] devised a system that utilizes wireless sensor networks in order to determine the position and trajectory of the mine fire danger, using a rating system. Muduli and Mishra [10] proposed a method dependent on a wireless sensor

mechanism for forecasting the emergence of underground mine fire danger. After modeling the suggested method and analyzing the design variables of mine fire prognosis, the machine learning-based monitoring tool was shown to be more dependable and accurate than the offline system. Tan and Wang [11] designed a wireless sensor network-based platform for mine safety that can monitor the environment in real time and issue alerts if a fire should break out.

With the discovery of spontaneous combustion of coal by Danish and Onder [12], underground mining safety might be guaranteed by early prediction of the fire. This action would protect miners from potential dangers in the mining industry process, such as fires. In order to obtain data for coal fire modeling and prediction, ten control points were set up. In terms of estimating fire severity, the fuzzy logic mechanism performed better than Graham's index. Numerical approaches have been found to be capable of analyzing procedures leading to ventilation of deep underground headings [13], and of assessing emergency situations related to the commencement of an oxidation reaction during optimal circumstances, which might result in self-ignition and self-heating. Extreme danger exists due to the concentration of coal spontaneous combustion and methane in experimental longwalls. These risks can put workers in a dangerous situation and cause significant delays in mining production [14]. The primary focus of rock engineering is the design and construction of structures resulting from mining production processes. To improve the safety and resiliency of production facilities resulting from underground mining processes, it is essential to conduct a comprehensive analysis of possible dangers.

There are a variety of data-driven techniques that can be utilized for predicting fire danger in underground mining engineering processes, enabling the invention of early warning mechanisms to take preventative actions in advance. However, the literature reveals that the implementation of innovative machine learning algorithms is still limited to predicting various fire danger situations in subterranean construction projects. A novel automation and Graham's ratio information management approach has been created to predict the probability of a mine fire in connection to environmental factors [15]. A hybrid framework based on both an artificial bee colony and a metabolic grey model was developed to predict coal stockpile burn conditions [16]. To promptly predict the passage of fire smoke in the tunnel, a system that incorporates simultaneous programming and data-driven techniques has been developed [17]. Correspondingly, numerous distinctive intelligence systems have been applied with the aim of solving the classification and regression problems of fires. A new methodology for predicting fire danger ratings levels in underground fires has been introduced, and it makes use of a combination of data-driven approaches [18]. Recently, a supervised machine learning technique including categorical boosting and a light gradient boosting algorithm have been applied to predict underground fire danger [19]. However, in a complicated phenomenon like an underground fire, supervised predictive models have significant limitations due to the difficulty of gathering an enormous number of high-quality labelled data. A combination with an unsupervised technique to improve the outcomes of a classification algorithm is an intriguing possibility for resolving this issue.

Researchers have achieved significant improvements in scientific or fundamental knowledge in recent years by employing novel methodologies, and this evidence has been shown to be directly applicable to safer production, encompassing environmental and sustainability problems. Various criteria for categorization are used to improve our understanding of established theories, methods, and laws regarding industrial production safety and danger evaluation [20]. It is well known that no one technological advancement can be construed as a strategic instrument for addressing the whole range of issues posed by dangerous sites [21]. Bayesian networks, with their malleable structure, can assess the safety of a wide range of danger situations [22]. The absolute objective of quality management is to maximize business profitability by minimizing production cycle expenditures. Concerns for the environment and public safety must be maintained while this is being carried out. Preventive maintenance planning might benefit from using danger assessment as a decision-making technique due to the fact that it can take into account all relevant factors,

including dependability, safety, and the environment. To lessen the likelihood of system failure and the consequences of such failure, maintenance planning should be guided by danger analysis (in terms of safety, economics, and the environment) [23]. Research was conducted to determine the level of danger caused by unplanned events including fires, explosions, and the discharge of poisonous chemicals [24]. When attempting to halt or reduce the negative consequences of a domino theory, decision-makers need a consistently reliable tool for developing realistic and optimal intentions [25]. Hydrocarbons, fire, explosion, and the spread of combustion products are all possible outcomes of an event in a sophisticated processing facility [26]. Even though there is rising support for implementing digital capabilities into system operation, there are significant safety issues that need to be resolved before the industry can go fully digital [27]. Breakthroughs in fundamental knowledge and the potential to develop safe production process, as well as the adoption of environmentally responsible practices in the mining engineering sector, in accordance with a set of specified standards, have been vital and significant.

Our level of scientific knowledge is insufficient to design underground production processes that are safe and environmentally friendly, and to be able to sustain pioneering solutions without rigorous testing. There is substantial proof linking these innovations to the technical field that improves the safety of mining production processes and preserves the environment. Moreover, the initiatives that aim to create fire danger rating for underground mining production processes have not yet integrated the application of state-of-the-art multi-criteria decision intelligence models. In this study, the isometric feature mapping (ISOMAP) technique would be employed with a fuzzy c-means clustering algorithm (FCM) and k-nearest-neighbors (KNN) algorithm to develop an early warning system for rating mine fires, persuading underground mining industry experts to adopt and support environmentally friendly mining industry processes.

The mechanism proposed in this research is an innovative and unused decision intelligence framework for identifying and assessing fire danger ratings. Its novel capabilities include a technique created as a result of observations of ventilation parameters, and a system that enables continual assessment of the mine fire danger and its prediction. In turn, the data-driven network's learning process enables the model to continually adapt to the underground mining production process in which it will be implemented. Therefore, this presents the prospect of extremely precise and speedy outcomes, both of which are crucial in the mining industry and in the situation of the fire danger.

The manuscript is organized in several sections. Section 2 presents a fire danger data acquisition system. In Section 3, a short review on the material and method used for the construction of the predicting mathematical model and performance indicators for evaluating the proposed mechanism are highlighted. Section 4 presents the results and discussion of the prediction of fire danger in underground mining production processes. Section 5 emphasizes the limitations of the developed and proposed models, followed by concluding remarks in Section 6.

## 2. Fire Danger Data Acquisition System

Researchers have considered temperature, oxygen, carbon dioxide, and carbon monoxide as input parameters [8,12] to improve their ability to predict the diagnosis and early recognition of fire danger in underground mining production processes. The key advantages of using these parameters are that it is easy to obtain their values, and they effectively portray the crucial conditions of fire classification. In this study, a large sample size (n = 120) of fire-related variables was gathered from Adularya coal mine in Turkey [12]. Mihalcck, where this mine is situated, is around 145 km from Ankara, Turkey's capital. This research project does not include a $CO_2$ assessment, since it is unlawful in Turkey to estimate the amount of $CO_2$ created during underground mining activities. Figure 2 displays the installations of the gas control center and the general ventilation system at Adularya coal mine. Gas samples are taken at eight different stations: the main intake, 510, 1410, 1409, 610/2B, 610, A06, and the main return, in addition to two stations in section D and D210 [12]. A

statistical description of the mine fire danger rating dataset in the proposed underground coal mine is shown in Table 1.

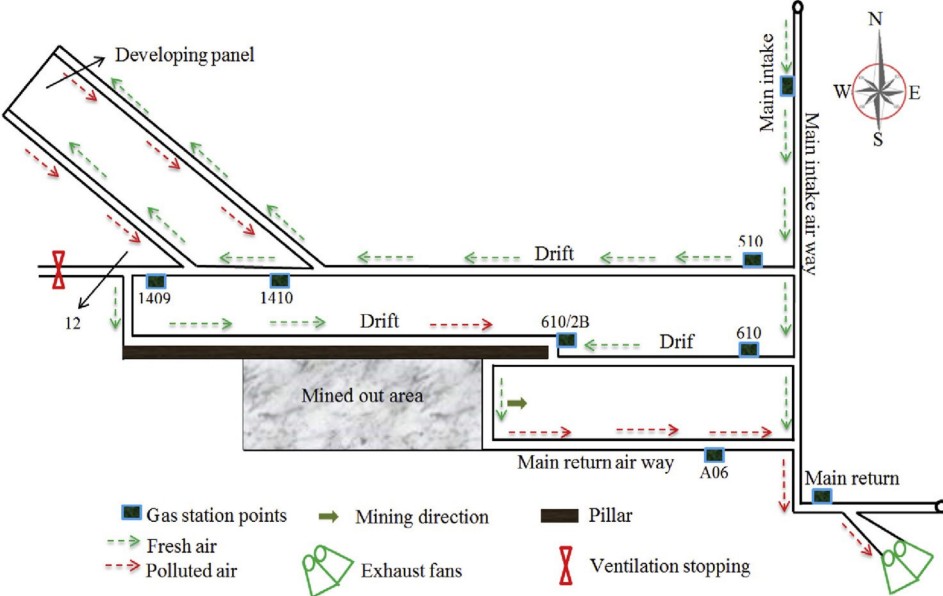

**Figure 2.** Installations of the gas control center and the general ventilation system [12].

**Table 1.** Statistical description of mine fire danger ratings influencing variables in underground coal mining.

| Parameters | Symbol | Unit | Minimum | Maximum | Mean | Standard Deviation |
|---|---|---|---|---|---|---|
| Oxygen | $O_2$ | % | 19.27 | 21.15 | 20.67 | 0.32 |
| Nitrogen | $N_2$ | % | 75.61 | 80.67 | 78.98 | 0.63 |
| Carbon Monoxide | CO | ppm | 0 | 6.31 | 1.19 | 1.51 |
| Temperature | T | °C | 0.68 | 29.03 | 21.47 | 6.13 |

## 3. Methodology

### 3.1. Isometric Feature Mapping (ISOMAP)

Isometric Feature Mapping (ISOMAP) is a non-linear procedure for reducing the number of features in a high dimensional dataset. It maintains the nonlinear elements of the initial data, which are generally eliminated in linear mechanism [28]. The ISOMAP exhibits nonlinear variations over a large domain, while maintaining linearity over smaller regions [29]. The term "multifaceted learning" is also sometimes used to describe this type of mechanism. Therefore, the manifold's local region is a metric space conversation [30]. The mechanism of ISOMAP is depicted in Figure 3. We use the symbols *a* through *f* to designate the appropriate datasets. Among their neighbors is a four-point star. The arc curve represents the actual location of the data points in the warp region. The green lines show the real distance in the database. It takes the original dataset and projects it onto a lower dimension, in the context of assuming that the high-dimensional records are representative of an area with a constant sample size. Then, this strategy attempts to uncover the concealed manifold [31]. Using either canonical Euclidean distance or any domain-specific metric, the geodesic distance may be estimated. To simulate domain-specific distance, one can use a sequence of preliminary steps between neighboring points in addition to the standard Euclidean metric of input data, which offers a decent simulation of geodesic distance [32]. ISOMAP integrates the essential algorithmic properties for effective computing, worldwide efficiency, and asymptotic resolution, while learning a wide range of nonlinear data points [33]. Table 2 shows the algorithm of ISOMAP. The

residual variance [34] is a suitable criterion for assessing ISOMAP's low dimensionality, which is illustrated as follows:

$$1 - R^2(D_M, D_G) \qquad (1)$$

whereas and $D_G$ depicts the geodesic distance and $D_M$ represents the eigen value that is derived by regenerating the ISOMAP geodesic distance function.

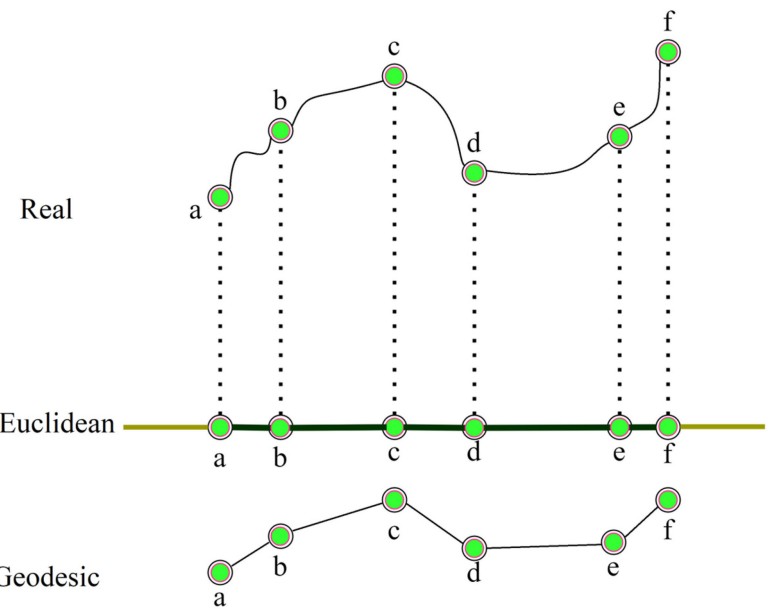

**Figure 3.** The concept of isometric feature mapping.

**Table 2.** Algorithm of isometric feature mapping employed in the study.

| ISOMAP Algorithm |
| --- |

**Step 1 Matrix construction**

    (a)    Squared pairwise similarities $D$, $D_{ij} = \left\| y_i - y_j \right\|^2$, distance space, computed on temporal axis. The *i,j* are special parameters

**Step 2 Graphing based on $D$**

    (b)    Build the weighted graph based on $D$, depending on each point's neighbors.

**Step 3 Estimating the geodesic distance**

    (c)    Estimate the geodesic distance $D_G$ by locating the weighted graph's optimal distance

**Step 4 Defining the values of $B$ and $J$**

    (d)    if $B = -\frac{1}{2J} * D_G J^T$ where $J = I - 1/N$ where $N$ represents the instances number and $I$ depicts the identity matrix.

**Step 5 Solving the eigen problem**

    (e)    The eigen problem $BP = P\sum$

**Step 6 Computing the principal vector**

    (f)    The major principal vectors are evaluated by $x = P\sum^{1/2}$

### 3.2. Fuzzy c-Means Clustering Algorithm (FCM)

A fuzzy c-means clustering algorithm (FCM) is a categorizing-based procedure that groups a data set into $N$ clusters, with each measurement contributing to each cluster to a specific extent. The paradigm of the fuzzy set was derived as the content validity of the issues of certainty and uncertainty in intelligent frameworks [35,36]. Multidimensional data may be clustered using the fuzzy logic concept, with each data point representing a percentage of a cluster. The method determines how close a sample is to a cluster's center, and uses that information to calculate the point's membership percentage. It is a form of unsupervised clustering that allows the creation of a fuzzy division inside the dataset. Each data point is assigned a degree of membership to each cluster based on its inverse distance from the cluster's center.

The FCM consists of four primary stages.

(i)　The initial stage requires the programmer to specify how many groups they want to analyze.

(ii)　A random cluster's centroid is used as the starting point for the method in the second stage.

(iii)　The cluster centroids are recomputed using the membership probabilities of the data points as the metric of choice.

(iv)　Until convergence is achieved, or a predefined maximum number of interactions is reached, the computation of centroids and update procedure will continue.

The Python fuzzy c-means module was used to determine the frequency of the mine fire's danger rating system in this study. The flowchart of FCM is illustrated in Figure 4.

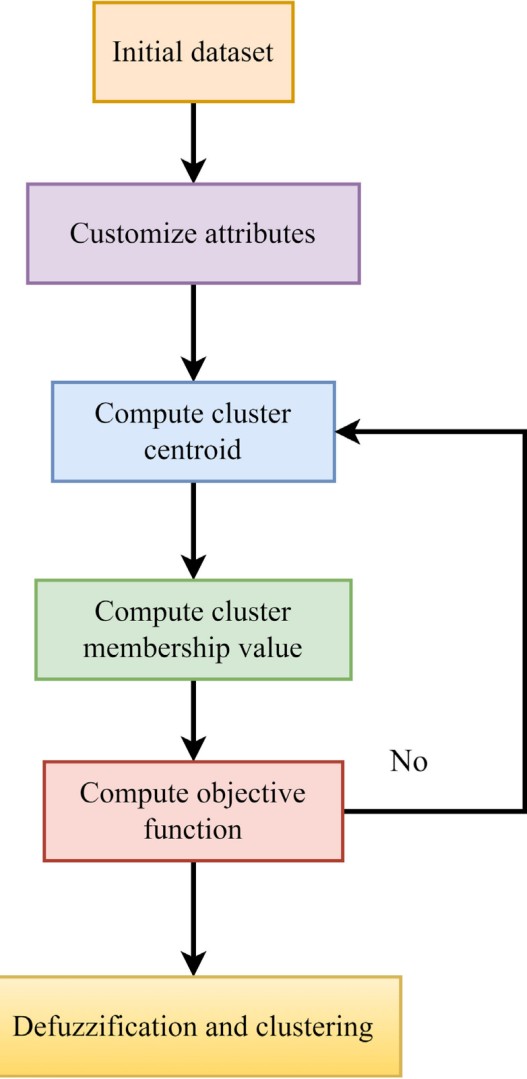

**Figure 4.** Various steps involved in fuzzy c-means clustering algorithm.

The FCM algorithm minimizes the optimized objective function, as illustrated in Equation (2), and operates as an optimized solution under the premise that the cluster number, "$c$", applies to the supplied mine fire database

$$J_{fcm}(C,D) = \sum_{k=1}^{c} \sum_{i=1}^{n} u_{ik}^{m} d_{ik}^{2} \qquad (2)$$

where $C = u_{ik}$ represents the membership vector of datapoint '$i$' in cluster '$k$', $D = \mu_1, \ldots \mu_c$ depicts the centroid set, and $d_{ik} = \|x_i - v_k\|$ illustrates the Euclidean distance between $x_i$ and cluster centre $v_k$. $u_{ik}$ fulfils the preceding Equation (3):

$$\sum_{i=1}^{n} u_{ik}^m = 1, \ whereas, \ i = 1, 2, 3 \ldots\ldots\ldots, n \tag{3}$$

The value $m$ is the cluster's overlapping indicator, known as the fuzziness index. In order to minimize the $J_{fcm}$, an estimated model of $C$ and $D$ can be obtained using an alternating selected optimization technique, as shown in Equation (4).

$$u_{ik} = \frac{1}{\sum_{i=0}^{n} \left(\frac{d_{ik}}{d_{ij}}\right)^{\frac{2}{m-1}}} \tag{4}$$

where $c$ and $i$ are integers in the domain between 1 and $n$, and $k$ is an integer in the domain between 1.

The cluster center $v_k$ can be computed using Equation (5)

$$v_k = \frac{\sum_{i=1}^{n} \left(u_{ik}^m x_i\right)}{\sum_{i=1}^{n} \left(u_{ik}^m\right)} \tag{5}$$

The FCM algorithm functions in five steps, as shown in Table 3.

**Table 3.** Fuzzy c-means clustering algorithm employed in the study.

| Fuzzy c-Means Algorithm |
| --- |
| 1   **Initialization** |
| 2   The values c, y and m are assigned |
| 3   Threshold values for convergence are determined |
| 4   $t := 0, TMAX := 50;$ |
| 5   $X := 0, TMAX := 50;$ |
| 6   $U^{(t)} := \{u_{11}, \ldots u_{ik}\};$ this is randomly generated |
| 7   **Compute the centroid** |
| 8   Compute the centroid $v_k$ |
| 9   **Perform classification** |
| 10   Computing and updating the membership matrix $U^{(t+1)} := \left\{u_{ij}\right\}$ |
| 11   **Prototype stabilization or convergence** |
| 12   if $t \leq TMAX :$ or $\left[abs\left(u_{ik}^t - u_{ik}^{t+1}\right)\right] \leq \varepsilon$ |
| 13       Iteration stops |
| 14   else: |
| 15   $U^{(t)} = U^{(t+1)} \ yt := t + 1;$ |
| 16   Iteration is repeated at (9) |
| 17   **Convergence is achieved** |

### 3.3. K-Nearest-Neighbors (KNN)

K-nearest-neighbors (KNN) is an appropriate classification technique that has proven extremely effective in practice. The KNN method relies on a voting scheme. It takes in information from a training database and applies it to future predictions about another database. The most recent advancement shows the effectiveness of KNN in minimizing distortion in a dataset [37–39]. The use of a KNN classifier for analysis of large datasets necessitates a substantial amount of processing power. The classification strategy uses the $k$ closest occurrences from the training sample to provide a label to a test sample. Establishing the association between every training sample and every testing sample is necessary for recognizing the $k$ nearest neighbors. Each data sample in the test set is given a home based on its $k$ nearest neighbors, as determined by the KNN algorithm. Finding the

*k* nearest neighbors requires computing the distance between all training as well as testing instances. Figure 5 shows the visual representation of k-nearest-neighbors. Table 4 depicts the architecture of KNN for the fire danger ratings dataset.

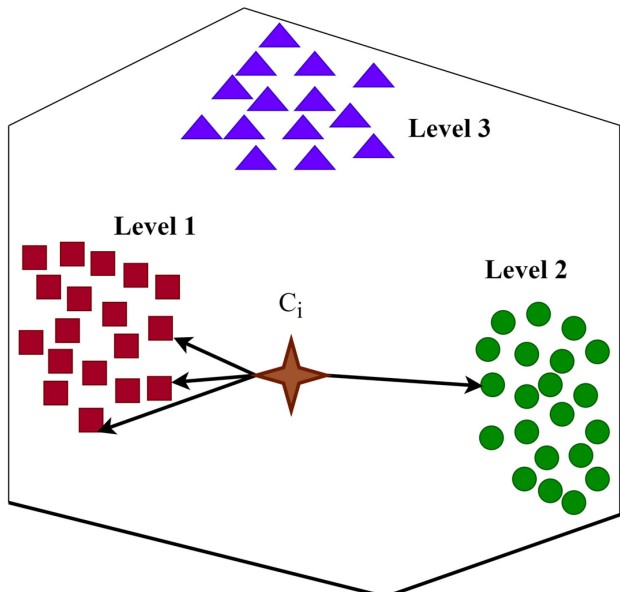

**Figure 5.** Visual demonstration of k-nearest-neighbors.

**Table 4.** Architecture of KNN for the fire danger ratings dataset.

---

**KNN Algorithm**

Input: A fire danger dataset and a number of test samples that need to be classified (the dataset has *t* dimension)

Output: The estimated fire danger rating in the testing dataset

(1)    Start

(2)    Utilizing the fuzzy c-means technique, divide the enormous amount of fire danger data into *m* distinct clusters.

(3)    $t = 1 \ldots m$ for each cluster.

(4)    Calculate the size of the cluster (number of data samples)

(5)    Compute $d_u^t$ $(u = 1, \ldots, t)$

(6)    For each test sample

(7)    Determine which data cluster is appropriate.

(8)    Apply the KNN algorithm on the chosen cluster to determine the test sample's estimated fire danger rating.

(9)    End

---

The KNN algorithm is given as follows:

### 3.4. Performance Indicators for Evaluating the Proposed Mechanism

For the objective of making predictions or sorting data into predefined levels, we may create data-driven or machine learning models for statistical classification systems. Due to the inadequacy of computer programmers, certain data points may be incorrectly classified. Researchers have employed a variety of performance assessment metrics, including precision, recall, F1-score and accuracy [40–42], to evaluate the feasibility of a classification-based data-driven model.

**Precision**

The potential of a classification technique to prevent data being classified as positive even when it is truly negative is referred to as Precision. It is expressed for each classification as the fraction of true positives relative to the total number of true positives and false positives.

$$Precision = \frac{TP}{TP + FP} \qquad (6)$$

### Recall

The recall or sensitivity of a predictor is its capacity to choose any effective occurrence. Recall is referred as the proportion of true positives to the sum of true positives and false negatives at each level.

$$Recall = \frac{TP}{TP + FN} \tag{7}$$

### F1-score

The F1 score is a weighted harmonic mean between 0.0 and 1.0 that depends on recall and accuracy. Due to the fact that F1-scores are derived by combining precision and recall, they often perform worse than accuracy measurements. When comparing classifier models, the weighted average of F1 is generally recommended as opposed to overall accuracy.

$$F1 - score = 2 \times \frac{Precision \times Recall}{Precision + Recall} \tag{8}$$

### Accuracy

Accuracy is determined by dividing the total sample size by the sum of true positives and true negatives. Providing the model is balanced, this is simply true. Inaccurate findings will result from a class imbalance.

$$Accuracy = \frac{TP + TN}{TP + FP + FN + TN} \tag{9}$$

The true positive (*TP*) level is the one that actually turns out to be positive, whereas the false positive (*FP*) level is the one that turns out to be negative; the false negative (*FN*) level is the one that turns out to be positive, and the true negative (*TN*) level is the one that actually turns out to be negative (which is predicted to be negative).

In addition, the confusion matrix is a prominent way to analyze data and resolve problems that require classification [43,44]. The primary diagonal line displays the total number of classes correctly predicted by the intelligent system. The greater the number of non-diagonal levels, the more frequently the intelligent system misclassified the corresponding set of characteristics. The ground truth level from the machine learning framework is displayed along the vertical axis of the confusion metric's plot, while the projected truth level is shown along the horizontal axis. As a result, the diagonal influences model accuracy more than any other factor. A simplified summary of the confusion matrix is illustrated in Figure 6.

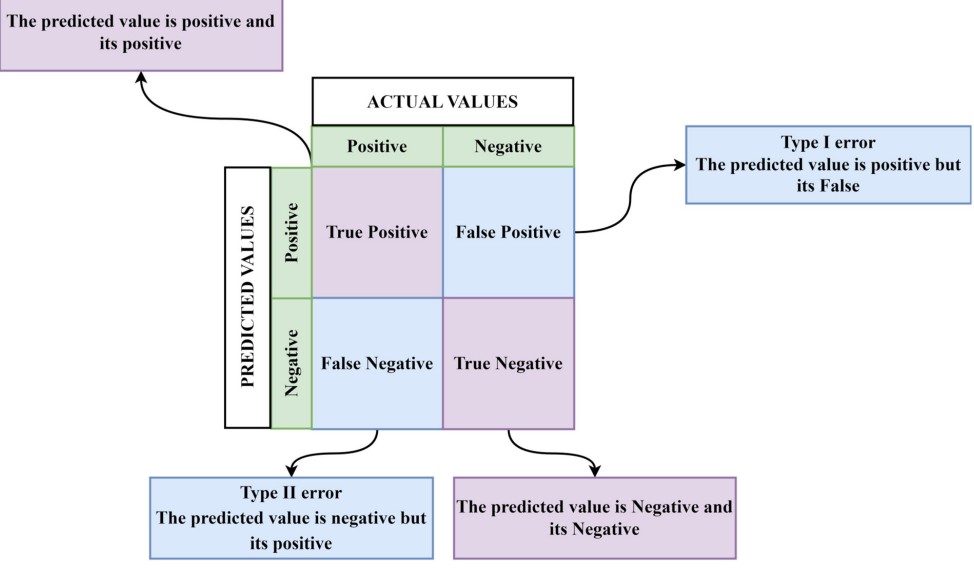

**Figure 6.** Brief overview of the confusion matrix.

## 4. Result and Discussion

Fire incidents in underground mining production processes are a major source of concern for millions of people across the world. Underground fires and explosions have continuously been the primary reason for a significant proportion of deaths and destruction of infrastructure through decades. The most typical causes of mine fires include inadequate ventilation, surface heat, electrical charges, and flame drivers in coal stockpiles located in airways aligned to the integrated ventilation system. In the most catastrophic scenarios, a mine fire might induce severe air contamination throughout the entire underground production process, resulting in the loss of several lives and the suspension of underground and construction operations. The expansion of air caused by a rapid increase in temperature, similar to that produced by an open fire, leads to the formation of combustion pollutants and vapor [26].

Decision intelligence is a realistic discipline that covers a vast array of decision-making strategies by integrating together many conventional and advanced disciplines to create, analyze, synchronize, implement, monitor, and adjust decision system and processes. A relatively new field, decision intelligence employs high-powered computing to steer, streamline, and automate process decisions. It helps businesses to analyze and forecast data in order to make more informed decisions at every management level, have more insight into their operations, and generate game-changing marketing benefits. Figure 7 portrays decision intelligence in a more comprehensive manner.

This study aims to employ a cutting-edge multi-criteria decision intelligence mechanism comprised of unsupervised and supervised machine learning approaches to effectively handle rock engineering challenges, with the ultimate goal of predicting and rating fire danger in underground mining production processes. In this research, we provide a three-stage process for predicting fire danger during underground mining operations:

(1) The magnification of the original fire database was reduced using an intuitive technique called ISOMAP.
(2) To minimize the consequences of sparse spectral heterogeneity in predominantly similar locations, the authors categorize the ISOMAP-acquired fire danger dataset using FCM.
(3) Lastly, a supervised machine learning algorithm known as k-nearest-neighbors is incorporated to predict different possible danger ratings of fire data in underground mining production operations. The flowchart of the proposed study is demonstrated in Figure 8.

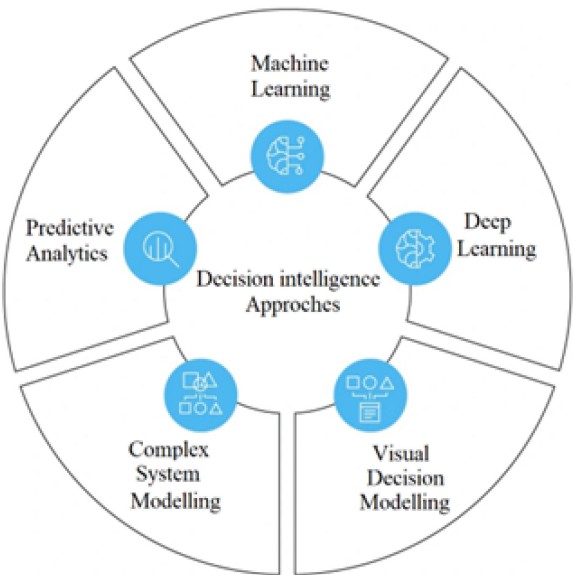

**Figure 7.** A deeper understanding of decision intelligence approaches.

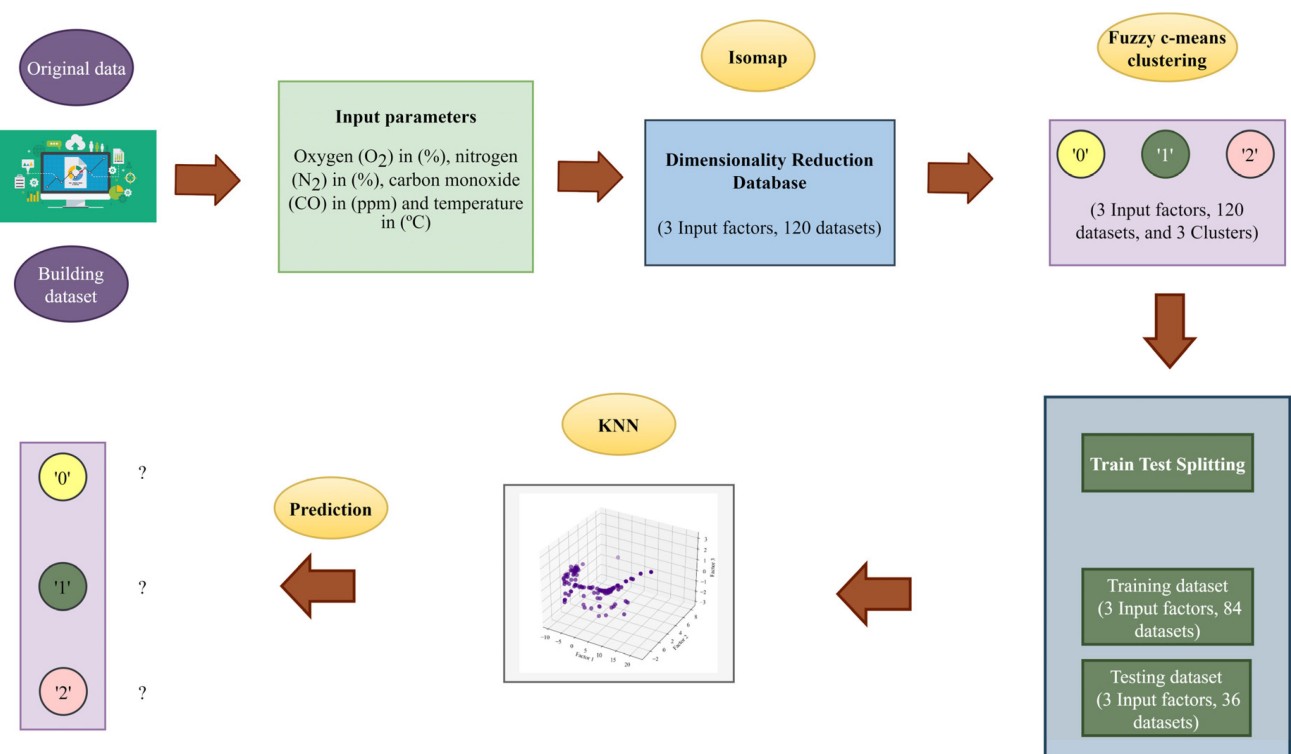

**Figure 8.** Flowchart of the proposed study.

Python is a globally known programming language that may assist in the exchange of conducting scientific and make the quantitative appraisal of huge databases significantly smoother. Python makes data mining tasks, such as clustering and classification, much easier to accomplish. This platform is among the finest operating tools for flexible application design, as it has numerous desirable features. As a conclusion, it is conceivable that the perspective of large-scale data processing might prove effective for the study of exceptionally vast quantities of fire-related data.

An implementation of the ISOMAP approach has been developed to generate a nonlinear manifold from fire datasets in underground mining production processes. It is based on an extension of the standard multidimensional scaling method of data dimensionality reduction. This procedure improves data recognition and enables us to assess variation between original data structures and the redeveloped spaces. The farness or closeness of the separation in the redeveloped spaces gives the magnitude of the resemblance between data points in contrast to the distances in the other traditional data dimensionality reduction constructed environments. The ISOMAP algorithm uses the geodesic distance to distinguish the separation between the attributes. Using many repetitions can help divide the data into manageable components that can then be clustered in a variety of ways. This technique's strength lies in its utilization of Dijkstra's method, which determines the shortest distance between two instances by traversing their neighbors in order to create distance estimations that more accurately reflect the variation in physical distance between data points. The ISOMAP module of the *sklearn.manifold* library has been implemented to execute the ISOMAP process. The original mining fire database is converted from a high-resolution matrix into a low-resolution matrix using the ISOMAP mechanism, as shown in Figure 9. For our analysis, we used the initial fire danger influence dataset, which includes information on four important parameters. The 3D reconstruction space of the fire danger data in underground engineering structures shows higher spatial variance than the original data points, which were equally spaced, as tabulated in Table 5.

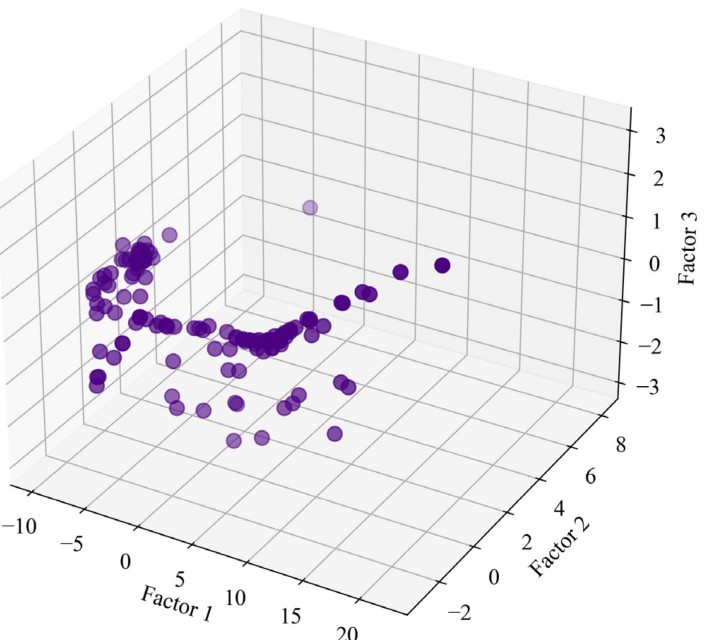

**Figure 9.** A three-dimensional structure of the ISOMAP-acquired fire danger dataset.

**Table 5.** Fire danger dataset following feature reduction using ISOMAP.

|  | Component 1 | Component 2 | Component 3 |
|---|---|---|---|
| **1** | 21.87642 | −0.61109 | 3.078009 |
| **2** | 18.05238 | −0.42372 | 2.588285 |
| **3** | 12.57939 | −0.12902 | 1.388911 |
| **4** | 11.55528 | 0.085894 | −1.79916 |
| **5** | 5.584876 | 0.243517 | −0.29477 |
| **. . .** | . . . | . . . | . . . |
| **116** | 2.666088 | 0.367277 | −0.36284 |
| **117** | 3.427989 | 0.335027 | −0.32689 |
| **118** | 5.010317 | 0.266068 | −0.19241 |
| **119** | 6.567639 | 0.189253 | 0.077908 |
| **120** | 7.433521 | 0.144162 | 0.250601 |

The majority of input features are in various scales and proportions; therefore, it is necessary to bring all features in identical scales and ranges. Scaling of features is used to accelerate and improve the computing process of machine learning techniques. As shown in Equation (10), a standardization technique has been employed for feature scaling in this study. Data standardization is a method for estimating one or more characteristics with a standard deviation of 1 and a mean value of 0 for each characteristic. In order to list the feature scaling, a standard scalar module from scikit-learn in the Python programming language has been chosen.

$$x_{std} = \frac{x - \mu}{\sigma} \tag{10}$$

where $x_{std}$ is the standard score, $x$ is the original value, $\mu$ represents the mean, and $\sigma$ is the standard deviation of a dataset's features.

Many problems encountered in analyzing geostatistical data may be addressed through the implementation of FCM. This tool may be used to design a process and fuzzy divisions from any kind of numerical data. These divisions are useful for providing evidence for already-established structures or for revealing hidden architecture in previously unknown data. In order to achieve the categorization of the fire danger characteristics dataset, FCM is employed, utilizing ISOMAP reconstructed points. The results of cluster monitoring have been recognized as generally applicable by the research community [45–47]. The silhouette

coefficient (or silhouette score) is a statistic used to evaluate clustering methods [48,49]. Its numerical value is between the range of −1 and 1. The value 1 indicates that the clusters are effectively differentiated and spatially separated, a score of 0 shows that the clusters are not differentiated or that no apparent variation exists between the clusters, and a score of −1 indicates that the clusters are allocated inaccurately in a dataset.

The silhouette coefficient may be used as proof that the standardized ISOMAP data have been properly categorized, because it reflects how the features have been organized according to the specific clusters. This index measures the effectiveness of the authentication employed via the clustering algorithm in selecting the best *k* cluster groups. Based on the existing literature [50,51], fire danger severity has been categorized into three primary ratings including low fire intensity (represented by 0), moderate fire intensity (represented by 1) and high fire intensity (represented by 2). As can be seen in Figure 10, the authors conducted a process with several iterations in the FCM analysis. The intensity of fire danger of designated by the colors yellow, aqua, and green. The silhouette score of 0.53 upon the sixtieth iteration of the *ISOMAP*-derived fire danger dataset proves that the clusters are reliable and consistent. The coordinates (−0.95, 0.59, 0.61), (−0.49, −0.94, 0.001) and (0.75, 0.17, −0.26) were the centers of the three clusters. A statistical evaluation of the performance reveals that the FCM can reliably categorize three distinct intensities of fire danger in underground mining production operations.

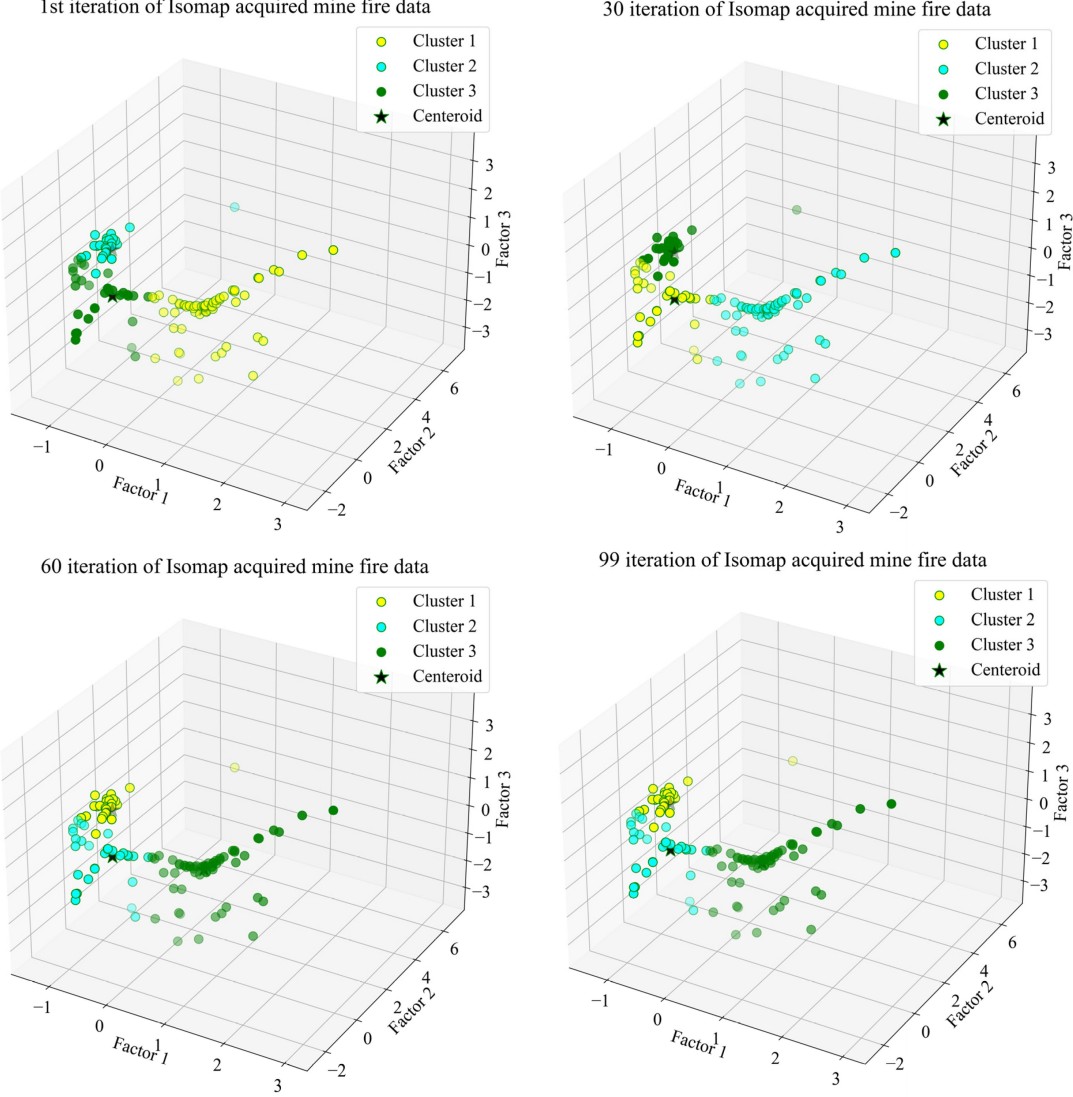

**Figure 10.** Several-iteration process in the FCM analysis.

The KNN is a supervised machine learning technique that is easy to implement and is useful for the classification of real-world problems. It has several potential uses in the field of rock engineering. To forecast an unknown fire danger intensity, it takes the *k* most comparable sample plots from the training dataset and applies the "feature similarity" concept to evaluate the observation according to how closely they match the predicted fire danger rating. In this study, the spectral distances were calculated by assigning a Euclidean distance from each unknown fire danger rating to each sample plot. Training data (consisting of approximately seventy percent of the FCM acquisition data) and testing data (consisting of approximately thirty percent of the FCM acquisition data) were chosen at random. The testing datasets are used to evaluate the framework's capacity to predict fire intensity ratings from previously unseen data in underground mining operations. The suggested KNN algorithm is developed using the default configuration of the Python program language. As shown in Figure 11, only two fire danger ratings are mispredicted in the testing datasets when using the application of the KNN machine learning approach when integrated with ISOMAP and FCM. The statistical results of the proposed mechanism are shown in Table 6.

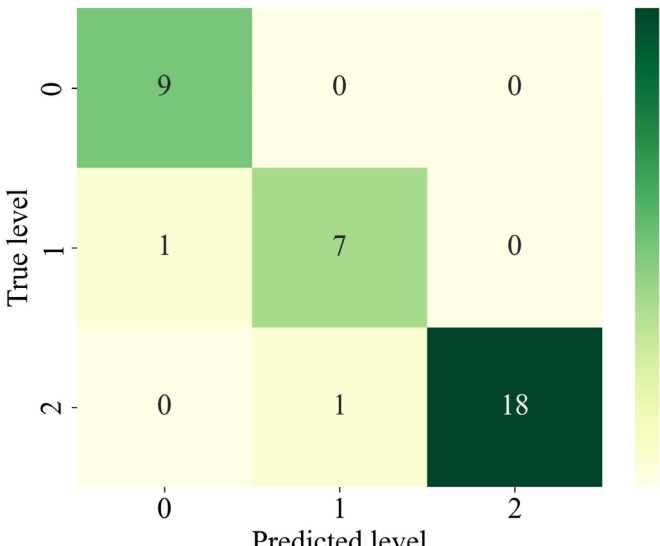

**Figure 11.** Confusion matrix of the k-nearest neighbor machine learning algorithm.

**Table 6.** Statistical results of performance indicators for evaluating the proposed mechanism.

|  | Low Fire Rating | Moderate Fire Rating | High Fire Rating |
|---|---|---|---|
| Precision (%) | 90 | 88 | 100 |
| Recall (%) | 88 | 88 | 88 |
| F1-score (%) | 100 | 95 | 97 |
| Proposed framework overall accuracy (%) | | | 94 |

The total accuracy of the KNN machine learning approach when integrated with ISOMAP and FCM is 94 percent. Previously, the authors were able to predict the potential for fire danger in underground mining production processes with 92 percent accuracy [19]. Consequently, the suggested intuitive decision-making mechanism has demonstrated its superior dependability and reliability over the previously published literature, and it may be utilized for the prevention and monitoring of fire danger in underground mining production processes.

Mine fires are often generated by coal ignition due to a pyrolysis action with ambient oxygen, but the associated dynamics driving this process are remain unclear. Due to the lack of a well-defined chemical composition, it is still unknown how coal and oxygen react

when subjected to moisture, steam, or fumes. Regardless of the cause, spontaneous fires in underground mining processes produce a large volume of hazardous and combustible gases, which can cause the destruction of underground mining machinery, the death of many workers, and the loss of significant coal resources. There is usually little chance of escape when a mine fire breaks out, increasing the number of casualties in underground engineering operations.

The risks of global warming, increasing temperatures, and carbon emissions can be mitigated if mining exploration is conducted in a safe, climate-smart, sustainable, environmentally friendly manner. This is because there is a global trend toward mineral-intensive, low-carbon advancements. A danger assessment system may assist emergency specialists in monitoring the activities of underground incidents, assessing their development over time, and responding to newly emerging fires. The proposed early warning system would aid emergency management professionals in monitoring the activities of mine fire danger ratings, following their movement over time, and responding to newly developing fires. The findings of this study provide a logical addition to the process of developing links between environmentally friendly mining production processes and sustainable technological advancement. This study aims to persuade underground mining industry experts to adopt and support safe mining industry processes that have not previously published in the existing studies.

The model presented for the prediction of fire danger in underground mining production systems offers a vast array of potential applications, which may be employed to ensure safe mining production and ventilation safety in the broadest sense. This applies not only to the procedures associated with its immediate practical application, but also to those methods. Automatic fire systems in mines require only a minimal amount of input from operators in the form of statistical information and calibration before a mining fire commences. As a result, they may be placed into operation very quickly. and even operate continually. When there is not enough time for an autonomous fire system to put out a blaze, it is crucial to provide access to measurement data as soon as possible. Although this will result in some delay in receiving results, the quality of those findings will not be compromised in any way. In general, the application that was built may be practically applied in a reasonably short amount of time, and without the need for any further processes. Access to the recorded data and the permission of the mine's managers are the sole requirements.

## 5. Limitation

Despite the fact that the suggested mechanism produces reliable predictions, the method that has been outlined has a number of shortcomings that will, at some point in the future, need to be addressed in order to be more effective.

(1) The dataset does not have a balanced representative sample. The accuracy of predictions made by machine learning algorithms is significantly impacted by a number of factors, including the quantity and quality of samples. When the dataset regarding an issue is relatively limited, the generalizability and dependability of a model tend to decrease. This is due to the fact that larger databases include more information that may be accessed. In addition, the dataset has a number of inconsistencies, most notably with the samples that include divergent values. This demonstrates the negative influence that inconsistent data may have on the outcomes. As a result, it is of the utmost importance to construct a database for the prediction of fires that is not only more extensive, but also more varied.

(2) The outcomes of the predictions might be affected by a variety of different indicators or attributes. Although the four attributes that were applied in this study were able to identify the essential fire situations to a certain limit, this does not indicate that other elements do not impact the fire prediction. As a consequence of this, it is essential to evaluate the effect that other important features have on the prediction results.

Fire safety is an approach that is sustainable. By reducing the frequency or severity of fires, it may be possible to avoid wastage of resources, lower pollution, and cut expenses. Whenever there is a fire in a mine, underground infrastructures are demolished, water is wasted, property is lost, and contaminants from the air are spilled into the surrounding environment. After a fire, any items that were destroyed are hauled to disposals and landfill, and clean resources are brought in to substitute them. When conducting an analysis of environmentally friendly production processes and sustainable practices, it is very necessary to take into consideration the estimated lifespan of the structure.

## 6. Conclusions

The subject matter that is discussed in this article has an enormous amount of significance, both from a theoretical but also from a practical perspective. This study proposed a multi-criteria decision intelligence framework to efficiently and precisely predict the danger of fire in underground engineering structures. The robustness of the developed framework was proved by evaluation of the results using multiple performance criteria. In this study, we utilized three commonly employed techniques in data science—ISOMAP, FCM, and KNN—to predict fire danger. Specifically, the data used in this study come from Adularya coal mine in Turkey. Several statistical performance indices are incorporated to evaluate the effectiveness of the fire hazard rating system in order to approximate an effective model for data prediction. The outcomes of the proposed model exhibit its capacity to produce highly accurate predictions of the fire danger. Therefore, it is advisable to implement the ISOMAP + FCM + KNN model as an accurate and appropriate model for predicting fire danger in underground engineering structures.

The developed research approach, which is based on the unsupervised–supervised classification algorithm, is an example of the practical use of advanced technological procedures to tackle a real-world problem. This research contributes a novel perspective on this issue by applying the results of the influencing parameters to the diagnosis and prediction of the rating level of fire danger using a data-driven intelligent mechanism. The difficulty of accurately estimating fire danger has significant real-world implications. The efficiency of this mechanism will have a critical influence on both the safety and the continuity of the underground mining production processes.

The most major application of the suggested approach is the prevention of fires in areas wherein hard coal is extracted by mining. This allows the underground mining production processes to continue operating in a safe and effective way. Even though they cost more, prevention strategies could prevent a fire from damaging the exploration region and causing severe financial losses for the mine due to the suspension of exploitation and isolation of the underground mining infrastructure.

In underground mining industry processes, the possibility of a fire must be considered extremely seriously. It is essential that underground structures undergo frequent evaluations of their fire safety in order to protect workers' lives and underground property from fire danger. There are several methods for preventing underground mining fire danger. By taking the necessary precautions, administrators of various facilities and workplaces where underground construction experts execute their duties can prevent these types of disasters. There have to be actions taken to guarantee workers' health and safety and to minimize mining industry financial losses as these sorts of events become more severe and frequent. As of now, despite some improvements, steps taken independently by mining corporations to combat mining fire danger have not been adequate. Therefore, this study seeks to determine whether or not miners are at risk due to fire danger. The proposed state-of-the-art decision-making paradigm is a major improvement because of its greater programming versatility and single-point monitoring. The intended design conforms to a stringent principle of zero occurrences. Underground projects are defined by the presence of a large number of people, or by the presence of valuable property; consequently, it is more challenging to evacuate in an emergency. Engineers and professionals that specialize in rock engineering create structures composed of rock that has been unearthed by mining

engineering. Understanding the dangers involved is essential for assuring the sustainable production and safety of facilities as a result of mining operations in the long term. The proposed multi-criteria decision intelligence permits early fire detection, providing the emergency response team with extra time to respond and extinguish the flames before they may spread, facilitating environmentally friendly technologies in underground mining production processes.

An attempt should be made to link the suggested model with the numerical model created using the data-driven intelligent approach. This is a highly significant topic that might be the subject of prospective future studies. This will make it feasible to identify in particular sites at which mine fires might occur. Since ventilation and other features of mine workings are now being continuously recorded by computerized devices, there is the potential to use the datasets that are produced as inputs for mathematical modelling of phenomena linked to fire and other ventilation dangers. By determining where additional measuring sensors should be installed, extending the investigation in this direction could have huge potential to promote green mining production, improve process safety, and increase the environmentally friendliness of operations. Analyses relying primarily on ventilation risks at post-mining waste sites and determining the amount of greenhouse gases emitted can be carried out with the suggested multi-criteria decision intelligence mechanism. As a result, the generated tool may be used in a variety of contexts, offering a vast number of opportunities for its use and enhancement in the sustainable, green, climate-smart, environmentally friendly and safe production of underground engineering processes.

**Author Contributions:** M.K.: Conceptualization, data curation, software, methodology, modelling, results, data collection, data analysis, coding, lead writing and visualization. M.K. and W.C.: writing—original draft. M.K., R.K.W., H.R. and D.A.M.: validation and formal analysis. M.K., R.K.W., H.R. and D.A.M.: supervision and project administration. Funding Acquisition: D.A.M. All authors have read and agreed to the published version of the manuscript.

**Funding:** The research was funded by the Ministry of Science and Higher Education of the Russian Federation (Project No. FSNM-2023-0005).

**Institutional Review Board Statement:** Not applicable.

**Informed Consent Statement:** Not applicable.

**Data Availability Statement:** The data used in this study is from published research: Danish and Onder [12] (https://doi.org/10.1016/j.shaw.2020.06.005, accessed on 23 August 2023). Additionally, the code supporting the findings of this study is available upon request from the corresponding author.

**Conflicts of Interest:** The authors declare no conflict of interest.

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
