# Peer review of "A Multi-Criteria Decision Intelligence Framework to Predict Fire Danger Ratings in Underground Engineering Structures"

_fire, doi:10.3390/fire6110412_

Round 1

Reviewer 1 Report

Comments and Suggestions for Authors

The objective of this study is to forecast the risk of fire in sub-surface mining production processes by the use advanced unsupervised and supervised machine learning methods. The effectiveness and superiority of the proposed multi-criteria decision intelligence mechanism have been demonstrated by its improved accuracy. This mechanism could possibly be effectively utilised for the prevention and monitoring of fire hazards in underground mining production operations. A major revision is suggested to improve the quality of the paper before it can be published on this journal. Here are the suggestions and comments that need to be addressed in the revision:

1) The authors are requested to include the result of the suggested method in the abstract. In addition, the method should also be elaborated in the abstract.

2) It is good to express the need for the study in the introduction. There is less background in this introduction stating the urge and novelty of the study in which innovative ideas must be flown through the background along with the useful insights. The literature review must be separate from the introduction section.

3) The overall performance accuracy difference between the suggested method and the method mentioned in literature algorithms is 3%. How is the performance accuracy of suggested method compared to the algorisms used in the literature?

4) Authors already discussed the limitations of the sample size and distribution of the dataset. Another major limitation of the dataset is that all the field data were collected from one coal mine. Please explain how the method can be applied to other coal mines and metal/nonmetal mines if the ISOMAP+FCM+KNN is trained by the field data from just one coal mine.

5) Oxygen is not a fire indicator - its amount decreases with the depth of the goaf.

Reviewer 2 Report

Comments and Suggestions for Authors

The paper mainly presents a multi-criteria decision intelligence framework for predicting the fire hazard in underground engineering structures and analyzes and discusses the prediction results of fire hazards in underground mining production. Fire hazard is predicted by utilizing supervised and unsupervised machine learning techniques. Isometric feature mapping is used to reduce the magnification of the original fire database. The fire hazard dataset acquired by the isometric feature mapping is categorized based on the fuzzy c-means clustering algorithm. The k-nearest-neighbors is incorporated to predict different possible hazard levels. Finally, Precision, recall, F1-score and accuracy are used to evaluate the feasibility. The method proposed in this paper provides a new approach for the monitoring and prevention of fire hazards in the underground mining production processes. Therefore, this paper can be published. However, there are some formatting, writing, and logical issues in the paper, as follows:

1.     "Underground" and "subsurface" appear many times in the article. What is the difference between them? It is recommended to think about the meaning of the two words;

2.     “Hazard levels”, “risk levels”, and “danger levels” appear many times in the article, and the meaning of these three expressions in the article seems to be the same. What is the difference between them? It is recommended to think about the meaning of them;

3.     In section 1, paragraph 4, the authors describe the researches on fire prediction to ensure the safe productions, while the latter describes that rock designing is largely concerned with the creation of rock structures that are the direct result of mining industrial processes. The context is incoherent, Moreover, what does rock designing refer to?

4.     In section 1, the authors discuss the relevant research comprehensively, but the content is too fragmented, the paragraphs are scattered, and the logic is not smooth enough. It is recommended that the authors refine the language and think the writing order and logic of this part;

5.     In section 3, the "Material" in the title is not described in the main text, is the title wrong here?

6.     In section 3.3, what is the classification criterion for the Fire Hazard Level? The basis for the classification of hazard levels is not given in the article;

7.     In section 3.4, The Brief overview of confusion matrix in Figure 6 is wrong;

8.     In section 4, The icon and content in Figure 7 do not seem to match;

9.     In section 4, In Figure 9, what is the coordinate value of factor 1, factor 2, and factor 3? How is the coordinate range determined? Figure 10 has the same problem. Moreover, why are the coordinate values different in Figure 9 and Figure 10?

10.   In Section 3.4, the authors use four metrics to evaluate the feasibility of the model, but in Figure 5, Only three indicators of achievement were counted;

11.   In Section 6, the author lacks a summary of the work done in this article, and overly describes the importance and significance of underground fire prediction. Such a conclusion is unreasonable.

Comments on the Quality of English Language

It is ok.

Round 2

Reviewer 1 Report

Comments and Suggestions for Authors

Authors have addressed my comments well. Now the paper is acceptable for publication.